

# Investigating Horizontal Axis Wind Turbine Aerodynamics Using Cascade Flows

Narges Golmirzaee and David H. Wood

Department of Mechanical and Manufacturing Engineering, University of Calgary, Calgary T2L 1Y6, AB, Canada
**Correspondence:** David H. Wood (dhwood@ucalgary.ca)

**Abstract.** The simplest aerodynamic model of horizontal-axis wind turbines is blade element-momentum theory. The blades are divided radially into small elements which are assumed to behave as airfoils when determining the lift and drag. Since all blades have neighbours, a more accurate two-dimensional representation is an infinite cascade of identical, equispaced lifting bodies. In this study, cascades of airfoils[1] at spacings and pitch angles typical of wind turbine applications, are analyzed using
the conventional and impulse forms of the force equations for two-dimensional, steady, incompressible flow. The flow at a Reynolds number of $6 \times 10^6$ through cascades of NACA 0012 airfoils was simulated using OpenFOAM software. The results of the force equations agree well (less than $1\,\%$ error) with the body forces determined directly from OpenFOAM for four spacing ratios. Examining the terms of these equations reveals the importance of the circulation, the viscous drag, and the displacement effect of the body's wake due to its finite width. We focus on the "wake vorticity" term, which is ignored in
blade element-momentum analysis. At a pitch angle of $90°$, this term balances the viscous drag when the angle of attack is zero. At zero pitch, which models the outer region of a wind turbine blade at high tip speed ratio, the term can account for $27\,\%$ of the axial thrust when angle of attack is about $4°$. This condition represents the rotor entering the high thrust region after the maximum power point. A simple equation is proposed for the wake vorticity term that is suitable for incorporation in blade element-momentum analysis. The normal force equation, like the angular momentum equation for wind turbines, has no
viscous term which forces the body drag to contribute to the circulation in the wake. It is shown that the airfoil assumption is conservative in that cascade elements always have higher lift:drag ratios than airfoils at the same angle of attack. An associated result is that separation occurs at higher angles of attack on a cascade element compared to an airfoil.

## 1   Introduction

An aerodynamic cascade consists of an infinite number of identical, equispaced bodies or elements, four of which are shown
in Fig. 1 (e.g., Wood, 2011). A cascade can thus be simulated as a single body with the remaining bodies represented by periodic boundary conditions, provided the width of the computational domain is the spacing between the elements, $S$. A cascade is the two-dimensional analogue of the blade element at any radius $r$ of a horizontal-axis wind turbine (HAWT), which is highlighted in red in part (b) of Fig. 1. The dashed lines show the annular streamtube intersecting the blade element, whose

---

[1]It may be verging on pedantry to note that an airfoil is an isolated body and so the phrase "cascade of airfoils" is an oxymoron. We will use "airfoil" to describe the shape of the bodies in the cascade.





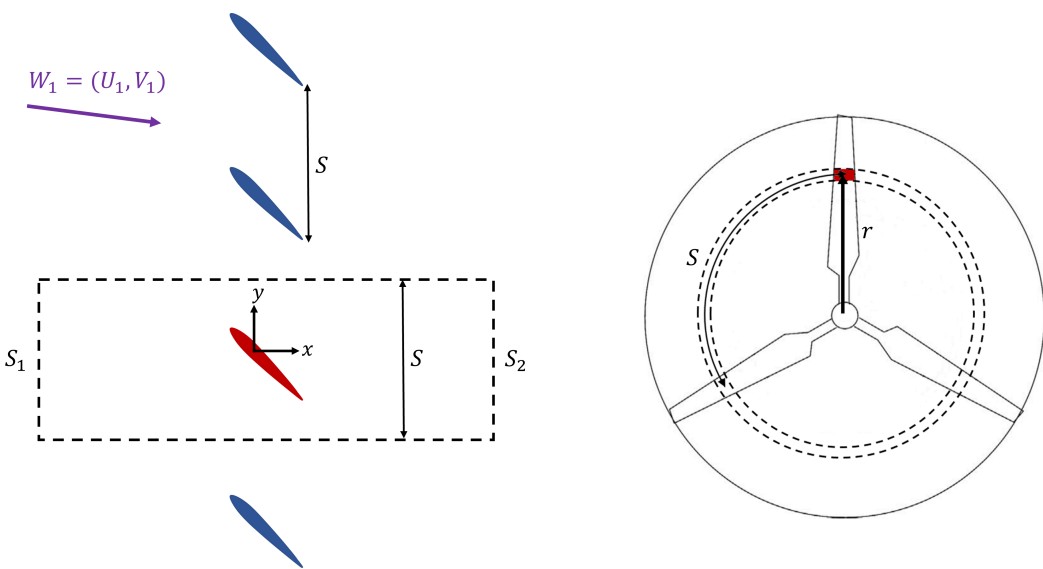

**Figure 1.** Two-dimensional equivalence between a cascade of infinite bodies and an annular section of a three-bladed HAWT at an arbitrary radial distance, $r$, from its rotational axis. The right figure is adopted from Hansen (2015).

axial and angular momentum change in response to the forces on the element. For a cascade, the analysis involves the $x-$ and $y-$direction momentum. Blade element-momentum theory (BEM) makes the critical assumption that the element forces are the lift and the drag on an airfoil of the same shape, Reynolds number, $Re$, and angle of attack, $\alpha$. As implied by the figure, this "airfoil assumption" is strictly valid only in the limit $S/c \rightarrow \infty$ where $c$ is the blade element or airfoil chord. There has not been, to the authors' knowledge, any independent test of the accuracy of the airfoil assumption.

The final geometric parameters for a cascade are the pitch angle, $\beta$, defined in Fig. 2, and the airfoil section. When the airfoil is operating at a small-to-moderate angles of attack, $\beta$ can be related to the tip speed ratio, $\lambda$, of a HAWT. Qualitatively, a large $\beta$ corresponds to a low $\lambda$, whereas the blade tips of modern HAWTs are often approximately parallel to the direction of rotation so that $\beta \approx 0$. When $\beta = 90°$ and the airfoil is symmetric, the only forces acting on the cascade are due to drag and we use this condition to check the derivation of the force equations. We thus chose to simulate a NACA 0012 section for $0 \leq \beta \leq 90°$.

The imposition of periodic boundary conditions to represent a cascade causes a component of the induced velocity (which can be thought of as generated by the vortex singularity representing the body) along the $y$-axis. The sum of this velocity in the $y$-direction, $\boldsymbol{V_{vor}}$, and the wind velocity, $\boldsymbol{W_1}$, is the velocity at the body, $\boldsymbol{W_{rel}}$, i.e., $\boldsymbol{V_{rel}} = \boldsymbol{V_1} + \boldsymbol{V_{vor}}$. $\boldsymbol{W_1}$ has a geometric angle of attack, $\alpha_g$, with the chord line. The angle between $\boldsymbol{W_1}$ and $\boldsymbol{W_{rel}}$ is called the induced angle of attack, $\alpha_i$. Also, $\alpha$ is the (effective) angle of attack, and it is defined as the angle between $\boldsymbol{W_{rel}}$ and the chord line (Fig. 2). Therefore, we have:

$$\alpha = \alpha_g - \alpha_i. \tag{1}$$





For isolated airfoils, $\alpha_g = \alpha$ as $\boldsymbol{V_{vor}} \to 0$ as $S/c$ becomes large. Using cascade analysis and the airfoil assumption, the lift, $\boldsymbol{L}$, and the drag, $\boldsymbol{D}$, exerted on a wind turbine blade element with the geometric angle of attack of $\alpha_g$ and the wind velocity of $\boldsymbol{W_1}$ is equal to $\boldsymbol{L}$ and $\boldsymbol{D}$ on an airfoil with the same shape but with $\alpha$ and $\boldsymbol{W_{rel}}$ (Wood, 2011). Figure 2 indicates that the angle between the force vector in the $x$-direction, $\boldsymbol{F_x}$, and $\boldsymbol{D}$ (or the force vector in the $y$-direction, $\boldsymbol{F_y}$, and $\boldsymbol{L}$), $\theta$, is given by:

$$\theta = 90 - \beta - \alpha. \tag{2}$$

$D$ and $L$ are the values of the drag and the lift per unit length, so the drag coefficient, $C_\mathrm{d}$, and the lift coefficient, $C_\mathrm{l}$, are defined as:

$$C_\mathrm{d} = \frac{2D}{\rho W_{rel}^2 c} \qquad \text{and} \qquad C_\mathrm{l} = \frac{2L}{\rho W_{rel}^2 c}. \tag{3}$$

The definitions of $C_\mathrm{d}$ and $C_\mathrm{l}$ here apply to cascade elements as well as airfoils. Once we know $\theta$ (or $\alpha$), $L$, and $D$, we have:

$$F_x = L \sin\theta + D \cos\theta, \tag{4}$$

and

$$F_y = L \cos\theta - D \sin\theta, \tag{5}$$

where $F_x$ and $F_y$ are the $x$ and $y$-components of the force per length on a blade element, respectively.

For a $B$-bladed HAWT at radius $r$, the spacing ratio is $S/c = 2\pi r/(Bc)$, which is the inverse of the local solidity, (Burton et al., 2011). For a conventional HAWT with three blades, $S/c$ lies between 1.5 and 40 (Table 3.3 of Manwell et al. (2009)) and

usually increases towards the tip. Apart from any effect on the airfoil assumption, there is another important consequence of this range of solidity: it is not possible to undertake wind tunnel tests of cascade models with solidities typical of modern wind turbines. It also turns out that cascade analysis reveals that the equations used in BEM thrust and torque are incomplete. The missing terms have cascade analogues whose importance can be determined from simulations in a straightforward manner.

One of the basic force equations in aerodynamics is the Kutta-Joukowsky equation for a steady, irrotational flow around an

airfoil (Hansen, 2015):

$$L = \rho W_{rel} \Gamma, \tag{6}$$

where $\rho$ is the density, and $\Gamma$ is the circulation around a closed curve, $C$, that encloses the body and its boundary layers. $\Gamma$ is given by

$$\Gamma = \oint_C \boldsymbol{W} \cdot \mathrm{d}\boldsymbol{l} = \oint_\Sigma \boldsymbol{\Omega} \cdot \mathrm{d}\boldsymbol{\sigma}, \tag{7}$$

where $\boldsymbol{W}$ is the velocity vector. $\Sigma$ is the surface whose boundary is $C$, $\mathrm{d}\boldsymbol{\sigma}$ is the local perpendicular unit vector on $\Sigma$, $\mathrm{d}\boldsymbol{l}$ is the unit tangent vector to $C$, and $\boldsymbol{\Omega}$ is the vorticity. The vorticity consists of the bound vorticity and the wake vorticity, but only the former contributes to the circulation. The Kutta-Joukowsky theorem relates the force to the vorticity. For a two-dimensional, steady, unbounded, inviscid flow, the drag is zero. Therefore, the Kutta-Jukowsky theorem for an airfoil indicates



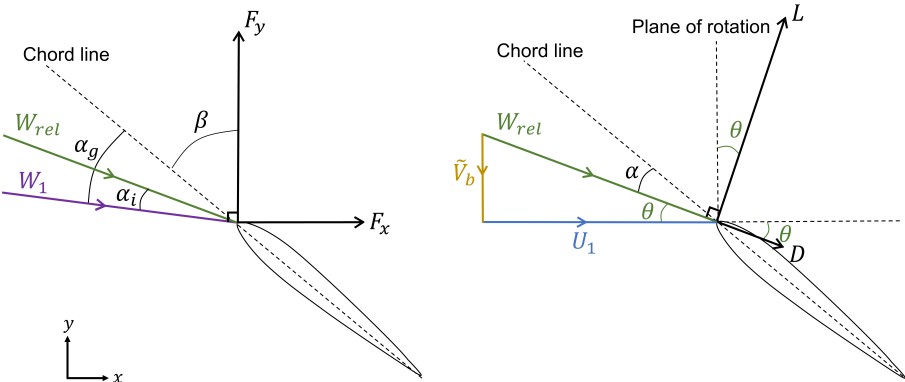

**Figure 2.** Representation of velocities, angles of attack, and forces on a cascade element. The plane of rotation of a blade element corresponding to this cascade element is parallel to the $y-$axis.

that the circulation does not contribute to the drag, or conversely, the drag does not affect the circulation. On this basis, many

implementations of BEM also ignore the contribution of the blade element drag to the circulation.

The conventional BEM equation for the thrust, which is equivalent to $F_x$, ignores a term associated with the vorticity in the wake. The vorticity in the body boundary layers and wake of cascade flows is in the transverse, $z$, direction. In contrast, a wind turbine wake has vorticity in all three directions associated with variations in the spanwise loading on the blades. Limacher and Wood (2021) accounted for this "shed" vorticity but not for the largely radial vorticity in the blade wake, which we will call

"wake vorticity" for both HAWTs and cascades. It is reasonable to expect that the wake vorticity is associated with element drag. Another important difference between shed and wake vorticity is that the former can move with the local streamlines and be force-free, as in Limacher and Wood (2021), whereas the latter cannot and this gives rise to a force. Limacher and Wood (2021) ignored the wake vorticity in deriving the Kutta-Joukowsky theorem for wind turbine thrust and determining the conditions for its application. We use the cascade version of their equation to assess the importance of the wake vorticity and

propose a simple equation for it that can be used in BEM analysis.

The main aims of the present study are to assess the accuracy of the airfoil assumption when the spacing ratio is finite, and the importance of four effects that are commonly ignored in the BEM equations for the streamtube intersecting the element. These are:

- the wake vorticity term as just defined,

- the quadratic terms in the nonlinear force equations. The axial and angular momentum equations in BEM, like the force equations derived below, are nonlinear. BEM effectively linearizes the equations and some effects of ignoring the wake in this process can be assessed from the cascade simulations.

- the blockage terms that may become important as the spacing ratio becomes small and the velocity deficit in the wake causes an increase in the velocity outside.



– the role of blade element drag in determining the circulation in the wake.

We also show that the airfoil assumption is conservative for angles of attack comparable to the angle giving the maximum
lift to drag ratio. That ratio is typically 10 % higher for the cascade we simulated than for a single airfoil, so we explored the
distribution of surface pressure and shear stress around the airfoil. We comment on the significance of the results in terms of
whether the finite solidity delays the stall on the cascade elements and, by implication, on blade elements.

The next section derives the forms of the force equations to be used in the OpenFOAM simulations described in Section 3.
The next section describes the airfoil results needed to assess the accuracy of the airfoil assumption. Section 5 describes the
cascade results and is followed by an assessment of the airfoil assumption. The last section contains the conclusions.

## 2    The conventional and impulse equations for cascade forces

Using conservation of momentum and the Reynolds transport theorem, the force, $\boldsymbol{F}$, on a stationary body submerged in a
two-dimensional, steady, incompressible flow can be written as (Noca (1997), Eq. (2.15)):

$$\frac{\boldsymbol{F}}{\rho} = \oint_{\Sigma} \hat{\boldsymbol{n}} \cdot (-\frac{p}{\rho}\boldsymbol{I} - \boldsymbol{W}\boldsymbol{W}) \mathrm{d}\boldsymbol{\sigma}, \tag{8}$$

where $\rho$, $\boldsymbol{W} = (U, V)$, and $p$ are the density, velocity vector, and pressure, respectively. $\Sigma$ is the external surface of the control
volume CV. The present analysis uses the rectangular, two-dimensional CV outlined by the dashed lines in Fig. 1 (a). $\hat{\boldsymbol{n}}$ is
the outward-facing unit normal on $S$. $\boldsymbol{I}$ is the $m \times m$ unit tensor, where $m = 2$ in this study, is the dimension of the space.
Using linear algebra techniques and the Navier-Stokes equation, Noca removed $p$ from the force equation to derive an impulse
equation (Eqs. (3.55) and (3.56) of Noca (1997)). For a stationary body in a two-dimensional, steady, incompressible flow, and
ignoring the viscous and other stresses at the CV boundary, these equations become:

$$\frac{\boldsymbol{F}}{\rho} = \oint_{\Sigma} \hat{\boldsymbol{n}} \cdot \big[\frac{1}{2}\boldsymbol{W}^2\boldsymbol{I} - \boldsymbol{W}\boldsymbol{W} - \boldsymbol{W}(\boldsymbol{x} \times \boldsymbol{\Omega})\big] \mathrm{d}\boldsymbol{\sigma}, \tag{9}$$

where $\boldsymbol{x} = (x, y)$ is position vector. This equation replaces the pressure by terms involving the kinetic energy and vorticity.
These two changes are important because the pressure contributions to wind turbine thrust are not easy to determine in general,
and the shed vorticity (as defined in Section 1) can be assumed to follow the local streamlines which simplifies the equation,
Limacher and Wood (2021). It is important to note that the impulse derivation gives the BEM thrust in terms only of the
circumferential velocity, equivalent to the $y$−direction velocity $V$ in Fig. 1(a), whereas the conventional BEM thrust equation
involves only the axial velocity, equivalent to the $x$−direction velocity $U$ in Fig. 1(a). This equation is derived by considering
the expansion of the flow through a HAWT which has no equivalent in cascade flow. The relationship between the two equations
for wind turbine thrust is discussed in Limacher and Wood (2021). Problems with pressure carry over to cascades where we
are interested in the equivalent of the radial vorticity in the blade wakes. Thus most of the subsequent analysis is based on Eq.
(9).





## 2.1 The forces on cascade elements

To derive the $x$ and $y$-components of the force on cascade elements, we used the rectangular CV in Fig. 1 (a) for all cases. Clearly, there can be no expansion of this two-dimensional flow, so the inlet face $S_1$ can be thought of as lying just upwind of a HAWT rotor and outlet face $S_2$ just downwind. There is only one lifting body in the CV, centred at $y = 0$. The $x-$direction boundaries are periodic, so any contribution to the momentum and vorticity fluxes along the top boundary cancels that of the bottom boundary in all derivations. Thus, the thrust and normal force are determined only by the fluxes through $S_1$ and $S_2$

and the only vorticity flux occurs at $S_2$. When $\beta$ is small, the wakes of the simulated and proximate bodies enter through the top and leave from the bottom boundary. Initially, we attempted to use CVs that avoided this crossing but these CVs had to be iterated and were difficult to implement. Later we show results that include wake crossings and argue from the accuracy of the force balances that allowing the crossings did not impair the numerical accuracy.

For such a CV, Eq. (8) becomes:

$$\frac{F_x}{\rho} = \int\limits_{S_1} (\frac{p}{\rho} + U^2)\mathrm{d}y - \int\limits_{S_2} (\frac{p}{\rho} + U^2)\mathrm{d}y \qquad (10)$$

and

$$\frac{F_y}{\rho} = \int\limits_{S_1} UV\mathrm{d}y - \int\limits_{S_2} UV\mathrm{d}y = U_1 V_1 S - \int\limits_{S_2} UV\mathrm{d}y. \qquad (11)$$

when the inlet velocity is $\boldsymbol{W_1} = (U_1, V_1)$. These two equations are the conventional force equations in the $x$ and $y$-directions for a cascade element. As noted above, the main difficulty in using them is the pressure at $S_2$ in Eq. (10).

At the inlet, the vorticity, $\Omega_z$, is zero, so the $x-$ and $y$-components of the force in Eq. (9) become:

$$\frac{F_x}{\rho} = \frac{1}{2}(U_1^2 - V_1^2)S - \int\limits_{S_2} \left(\frac{1}{2}U^2 - \frac{1}{2}V^2 + yU\Omega_z\right)\mathrm{d}y \qquad (12)$$

and

$$\frac{F_y}{\rho} = U_1 V_1 S - \int\limits_{S_2} \left(UV - xU\Omega_z\right)\mathrm{d}y. \qquad (13)$$

From now on, we denote the $x$ and $y$-components of the velocity vector at $S_2$ by $U_2$ and $V_2$, respectively, and consider first

the further development of Eq. (12). We note in passing that the last term in the $S_2$ integral is the wake vorticity term as $\Omega_z$ is non-zero only in the element wake. By moving $S_2$ far enough downstream, we can apply the slender flow approximation $\Omega_z \approx \partial U_2 / \partial y$. Using conservation of mass, the approximate form of (12) becomes:

$$\frac{F_x}{\rho} \approx \frac{1}{2}\left(U_1^2 - \overline{U}_2^2\right)S + \int\limits_{wake} \left(\overline{U}_2^2 - U_2^2\right)\mathrm{d}y - \widetilde{V}_b \Gamma + \frac{1}{2}\int\limits_{S_2} v_2^2 \mathrm{d}y, \qquad (14)$$

where $\overline{U}_2$ is the value of $U_2$ outside the wake. The subscript $wake$ on the first integral indicates that the integrand is non-zero

only within the element wake at $S_2$. $\widetilde{V}_b = (V_1 + \widetilde{V}_2)/2$ is the average normal velocity at the body, where $\widetilde{V}_2$ is the average of





$V_2$. $\Gamma$ is the bound circulation of the body, and $v_2 = V_2 - \widetilde{V}_2$. Each of the four terms on the right-hand side of Eq. (14) has an interpretation that applies at any distance from the cascade and it will be shown that the magnitude of the individual terms does not change substantially with $x$. The first two terms represent the displacement effect of the wake and the increase in the velocity outside the wake due to its finite thickness. The third term can be related to the $x$-component of the lift on the

body. It is one component of the Kutta-Joukowsky theorem for the lift on a single airfoil, and is the contribution of the bound circulation to the thrust. It is similar to the induced drag in the lifting line theory of wings. The fourth term arises from any non-uniformity of $V_2$ over the outlet and the non-linearity of the thrust equation. Limacher and Wood (2021) ignored the blade drag and assumed a rotor with infinitely many blades with no circumferential variation in the velocities. If their analysis is applied to a cascade, only the third term remains. One of the important applications of the present analyses is to understand the

importance of the terms that were ignored.

Applying conservation of mass between $S_1$ and $S_2$ results in:

$$U_1 S = \overline{U}_2(S - \delta) + \int\limits_{wake} U_2 \mathrm{d}y, \qquad (15)$$

where $\delta$ is the thickness of the body wake at $S_2$ whose precise definition is not important. Eq. (15) can be rewritten using:

$$(U_1 - \overline{U}_2)S = \int\limits_{wake} (U_2 - \overline{U}_2)\mathrm{d}y \qquad (16)$$

or

$$\frac{1}{2}\left(U_1^2 - \overline{U}_2^2\right)S = \frac{1}{2}\int\limits_{wake} (U_1 + \overline{U}_2)(U_2 - \overline{U}_2)\mathrm{d}y, \qquad (17)$$

Using Eq. (17), Eq. (14) becomes:

$$\frac{F_x}{\rho} \approx \int\limits_{wake} \left(U_2 + \frac{\overline{U}_2 - U_1}{2}\right)\left(\overline{U}_2 - U_2\right)\mathrm{d}y - \widetilde{V}_b\Gamma + \frac{1}{2}\int\limits_{S_2} v_2^2 \mathrm{d}y. \qquad (18)$$

At least in some circumstances, the first term can be associated to the $x$-component of the drag on the body. In Section 5.1, it

is shown that as $S/c \to \infty$ this term reduces to the drag equation for a single non-lifting body.

To develop Eq. (13), we use Eqs. (1.5) and (1.6) of Liu et al. (2015) which are combined as Eq. (9.1.20) of Wu et al. (2015):

$$\int\limits_{S_2} U\Omega_z \mathrm{d}y = 0 \qquad \text{and} \qquad \int\limits_{S_2} \Omega_z \mathrm{d}y = 0. \qquad (19)$$

These equations were derived for airfoil flow with the wake normal to the equivalent of $S_2$ in our study, and placed where the slender flow approximation is valid. Then the airfoil drag is given entirely by the wake vorticity term – see Eq. (1.8) of Liu

et al. (2015) – which is the last one in Eq. (12). Liu et al. (2015) call this term "Taylor's drag formula" in reference to Taylor's (1926) derivation in the appendix to Bryant and Williams (1926). The restrictions on the wake apply to many of the CVs used here, but not when the blade pitch and angle of attack are small as in the tip region of wind turbine blades at high tip speed





ratios. These conditions cause the wake crossings as mentioned above, and the wake to leave the $CV$ at a small angle to $S_2$ as documented below. The first equation, which prevents any flux of vorticity from $S_2$, is always valid for our simulations and is

the more important as it removes the $\Omega_z-$term from the $x-$direction force in Eq. (13) and makes the conventional equation for $F_y$ identical to the impulse one. Validity is easy to show using the principle that $F_x$ cannot depend on the origin for $y$. The status of the second equation, which removes the wake vorticity from the determination of the circulation, is more problematic but less important. Equation (13) becomes:

$$\frac{F_y}{\rho} = U_1\Gamma - \int_{S_2} u_2 v_2 \mathrm{d}y, \tag{20}$$

where $u_2 = U_2 - \widetilde{U}_2$ and $\widetilde{U}_2$ is the average of $U_2$. From Eq. (5), $D$ contributes to $F_y$, which has only a circulation and a quadratic term, which is shown later to be very small. Therefore, viscous effects must influence the circulation whenever the element drag has a $y-$component.

If we ignore the vorticity in the wake and spatial variations in the velocities, Eqs. (14) and (20) and the velocity at the body approximately become:

$$\frac{F_x}{\rho} \approx -\widetilde{V}_b\Gamma, \qquad \frac{F_y}{\rho} \approx U_1\Gamma \qquad \text{and} \qquad \boldsymbol{W_{rel}} \approx (U_1, \widetilde{V}_b). \tag{21}$$

From Eqs. (4), (5), and (21), we have:

$$L = F_y\cos\theta + F_x\sin\theta \approx \rho U_1\Gamma\frac{U_1}{W_{rel}} + \rho\widetilde{V}_b\Gamma\frac{\widetilde{V}_b}{W_{rel}} \approx \frac{\rho W_{rel}^2\Gamma}{W_{rel}} = \rho W_{rel}\Gamma \xrightarrow[S/c\to\infty]{} \rho W_1\Gamma, \tag{22}$$

which is the Kutta-Joukowsky equation for an airfoil.

For later use, we note that $F_x/\rho$ and $F_y/\rho$ can be normalized by $W_1^2 c/2$:

$$C_\mathrm{x} = \frac{2F_x}{\rho W_1^2 c} \qquad \text{and} \qquad C_\mathrm{y} = \frac{2F_y}{\rho W_1^2 c}. \tag{23}$$

## 3   OpenFOAM simulations

Since it is not possible to experimentally measure cascade forces accurately over the range of $S/c$ for a typical HAWT, the flow was simulated using OpenFOAM (Greenshields, 2021), which is a well-known open-source CFD software. OpenFOAM calculates the values of $F_x$ and $F_y$ directly from the pressure and the shear stress on the surface of the body.

This study used the Spalart-Allmaras model, one of the most common RANS turbulence models (Spalart and Allmaras, 1994), and the $k-\omega$ SST (Menter, 1993) model. The Spalart-Allmaras model is a one-equation model that is accurate for airfoil flows in terms of the mean velocity distribution in the wake (Thomas and Liu, 2004). The simulations were done for steady, incompressible flows, so we chose the simpleFoam algorithm, which is used for incompressible, steady, turbulent flows. In the present work, the flow around an isolated NACA 0012 airfoil with $\alpha = 0°$, $5°$, $7°$, and $10°$ was simulated, followed by

the cascade simulations using the same airfoil. Most simulations were done with the Spalart-Allmaras model and only a few with the $k-\omega$ SST for comparison. In the following, any unspecified result used the former model. All $k-\omega$ SST results are indicated as such.





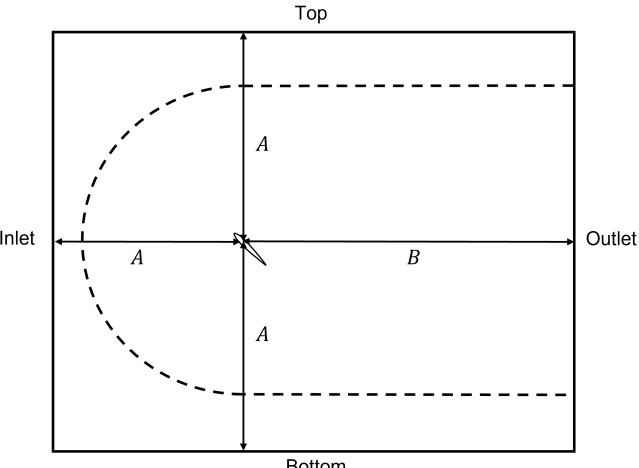

**Figure 3.** Domain sizes and boundary names for the airfoil simulations with $c = 1$. The aerodynamic centre of the airfoil is at the end of the arrow from the inlet. The dashed lines show a typical C-grid used for many airfoil simulations.

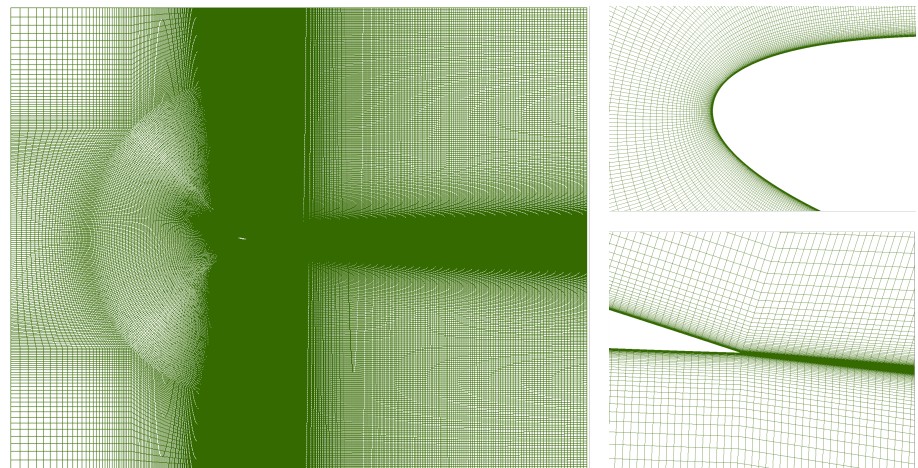

**Figure 4.** Far view and near view of the mesh implemented in the simulation of NACA 0012 airfoil at $\alpha_g = 10°$.

### 3.1 Boundary conditions and domain size for the airfoil simulations

$Re$ was set to $6 \times 10^6$ to have turbulent flow over most of the airfoil to avoid issues with simulating transition. Based on Michna
et al. (2021), ignoring of transition will cause the predicted drag to be high even at this high $Re$. The freestream velocity, $W_1$
and the kinematic viscosity, $\nu$ are set to $51.48 \ ms^{-1}$ and $8.58 \times 10^{-6} \ m^2 s^{-1}$. Many airfoil simulations use a C-grid (e.g.,
Eleni et al., 2012), but we used a rectangular domain as shown in Figs. 3 and 4. The reason is that cascade simulations must
separate the inlet from the top and the bottom boundaries to allow the imposition of different boundary conditions. Versteeg and





**Table 1.** Boundary conditions for single airfoil simulations using Spalart-Allmaras model.

| Boundaries | $U\ [ms^{-1}]$ | $p\ [m^2 s^{-2}]$ | $\nu_t\ [m^2 s^{-1}]$ | $\tilde{\nu}\ [m^2 s^{-1}]$ |
|---|---|---|---|---|
| Inlet | 51.48 | zeroGradient | $8.58 \times 10^{-6}$ | $3.432 \times 10^{-5}$ |
| Airfoil | 0 | zeroGradient | 0 | 0 |
| Outlet | zeroGradient | 0 | zeroGradient | zeroGradient |

**Table 2.** Boundary conditions for single airfoil simulations with the finest grid (first grid distance of $2.7 \times 10^{-7}\ c$) using $k - \omega$ SST models.

| Boundaries | $U\ [ms^{-1}]$ | $p\ [m^2 s^{-2}]$ | $k\ [m^2 s^{-2}]$ | $\nu_t\ [m^2 s^{-1}]$ | $\omega\ [s^{-1}]$ |
|---|---|---|---|---|---|
| Inlet | 51.48 | zeroGradient | $8.58 \times 10^{-8}$ | $8.58 \times 10^{-9}$ | 10 |
| Airfoil | 0 | zeroGradient | 0 | 0 | $9.416 \times 10^{10}$ |
| Outlet | zeroGradient | 0 | zeroGradient | zeroGradient | zeroGradient |

Malalasekera (2007) suggest using Dirichlet boundary conditions for all variables except for pressure and Neumann boundary

condition for pressure at the inlet. At the outlet, specified pressure and zero gradient boundary conditions for other variables are recommended (Versteeg and Malalasekera, 2007). The boundary conditions for the left boundary are used for the top and bottom boundaries (Versteeg and Malalasekera, 2007). Tables 1 and 2 show the boundary conditions used for Spalart-Allmaras and $k - \omega$ SST simulations. The values are within the ranges recommended by Spalart and Rumsey (2007), Spalart (2000) and Menter (1994). The aerodynamic centres of the airfoils and bodies were placed at the origin of the co-ordinate system. For

the airfoil simulations with $c = 1$, the distances from the inlet, top and bottom boundaries to the origin are denoted by $A$, and the distance between the origin and the outlet boundary is denoted by $B$ (Fig. 3). For the cascade simulations, $A = S/2c$, and the choice of $A$ and $B$ is described below. To accurately resolve the vorticity field in the regions of large spatial gradients, the meshes in the wake of bodies are denser (Fig. 4).

  Rahimi et al. (2014) propose a domain size of $A = 25$ and $B = 30$, and Eleni et al. (2012) recommend $A = 20$. Therefore,

the initial value for the domain size for the single airfoil simulations was $A = 20$ and $B = 30$. First, for the airfoil at $\alpha = 10°$, we fixed $B$ and changed $A$ to decide the inlet size. After choosing the appropriate value for $A$, we changed $B$. Table 3 presents the percentage difference of $C_d$ and $C_l$ between all cases and the case with $A = 20$ and $B = 30$. The percentage difference between two numbers $a$ and $b$ is defined as $100\ \% \times 2|a - b|/(a + b)$.

  The first domain size, $A = 20$ and $B = 30$, was chosen as it has the minimum number of cells and gave less than 1 %

difference in $C_d$ and $C_l$ from the bigger domains. For this grid, the variations in $C_d$ and $C_l$ vs number of cells are shown in Fig. 5. The coefficients vary monotonically as expected if the number of cells is in the region where Richardson extrapolation is valid. Details of the full grid convergence study are omitted for brevity. In summary: the value of $\mathrm{GCI}_{23}/\mathfrak{r}^p \mathrm{GCI}_{12}$, as defined in Roache (1994) , for $C_d$ and $C_l$ are both very close to 1, which is further evidence that they are in the asymptotic range of





**Table 3.** Domain independence study for the simulation of the single airfoil at $\alpha = 10°$: $C_\mathrm{d}$ and $C_\mathrm{l}$ for the airfoil with $\alpha = 10°$.

| Case No. | $A$ | $B$ | $C_\mathrm{d}$ | % difference of $C_\mathrm{d}$ with case 1 | $C_\mathrm{l}$ | % difference of $C_\mathrm{l}$ with case 1 |
|---|---|---|---|---|---|---|
| 1 | 20 | 30 | 0.01287 | 0.00 % | 1.07379 | 0.00 % |
| 2 | 10 | 30 | 0.01335 | 3.66 % | 1.07291 | 0.08 % |
| 3 | 30 | 30 | 0.01294 | 0.54 % | 1.07298 | 0.08 % |
| 4 | 20 | 20 | 0.01333 | 3.51 % | 1.07103 | 0.26 % |
| 5 | 20 | 40 | 0.01277 | 0.78 % | 1.07436 | 0.05 % |

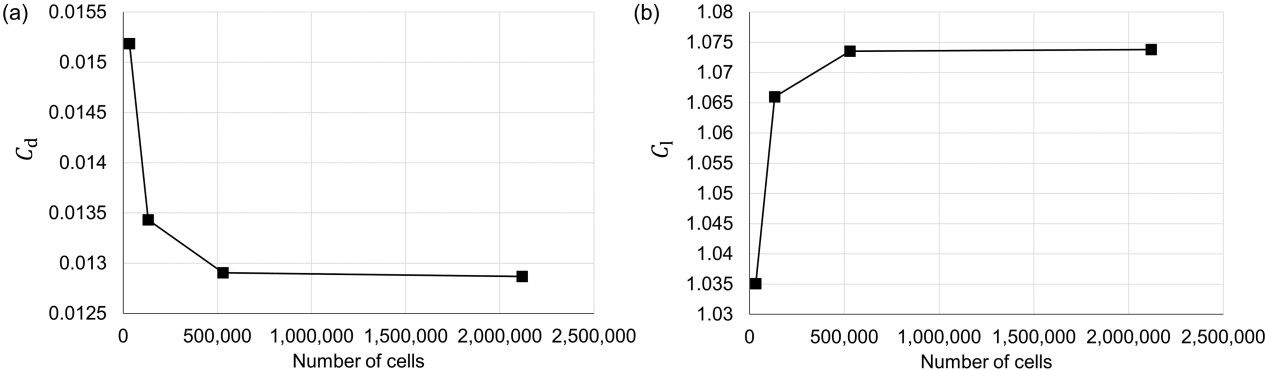

**Figure 5.** Grid convergence study of (a) $C_\mathrm{d}$ and (b) $C_\mathrm{l}$ for single airfoil simulation with $\alpha = 10°$ and $A = 20$ and $B = 30$.

convergence. The estimated $C_\mathrm{d}$ and $C_\mathrm{l}$ at zero grid spacing are 0.01287 and 1.07380 with numerical uncertainty bounds of 0.03 % and 0.001 %, respectively.

It turned out that the cascade simulations needed separate checks on domain size and grid number, so we take the unusual step of describing these prior to presenting the cascade results below.

### 3.2 Dimensionless distance from the wall ($y^+$)

Simulations of turbulent wall-bounded flows require a large number of cells to accurately resolve the velocity gradients near the wall. The most important parameter here is the dimensionless distance from the wall to the first cell centre, $y^+$, defined as (Schlichting and Gersten, 2017):

$$y^+ = \frac{yu_\tau}{\nu} \tag{24}$$

where $y$ is the distance from the wall, $\nu$ is the kinematic viscosity and $u_\tau$ is the friction velocity. The convention in CFD is that the cells touching the surface have $y^+ < 1$ to ensure the validity of the universal relation $u^+ = u/u_\tau = y^+$.

The recommended $y^+$ value to simulate an incompressible steady state flow at high Reynolds number using the Spalart-Allmaras model is $y^+ < 1$, (Eça et al., 2018). Also, based on Eça et al. (2018), the recommended $y^+$ value for $k - \omega$ SST is





**Table 4.** $C_\mathrm{d}$ and $C_\mathrm{l}$ for different grids for the simulation of single airfoil at $\alpha = 10°$, $A = 20$ and $B = 30$.

| Grids | Cell numbers | $C_\mathrm{d}$ | $C_\mathrm{l}$ |
|---|---|---|---|
| Coarsest grid | $33,112$ | 0.01519 | 1.03507 |
| Coarse grid | $132,448$ | 0.01343 | 1.06598 |
| Medium grid | $529,792$ | 0.01291 | 1.07352 |
| Fine grid | $2,119,168$ | 0.01287 | 1.07379 |

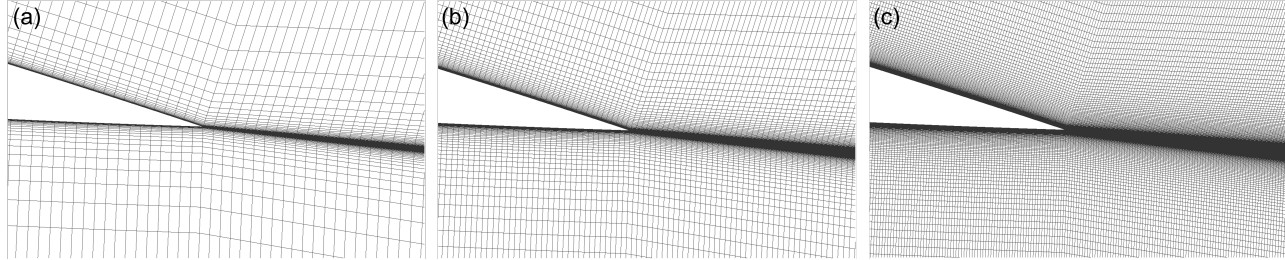

**Figure 6.** Grids at the trailing edge for (a) coarse, (b) medium, and (c) fine.

$\simeq 0.1$, so the finest grid, which has around 2 million cells and the maximum $y^+$ of $\simeq 0.1$, was used. Table 5 shows the distance of the first grid from the airfoil and $y^+$ of the coarse, medium and fine grids. For each grid, the first cell had the same height everywhere on the airfoil.

## 4 Single airfoil results

In this section, $C_\mathrm{d}$ and $C_\mathrm{l}$ of the single airfoil simulations at $\alpha = 10°$, $7°$, $5°$, and $0°$ are presented and compared with the experimental data from Ladson (1988). For all these angles, the domain size and the grid number that have been already employed for the airfoil at $\alpha = 10°$ were used.

The experimental data are for NACA 0012 at $Re \approx 6 \times 10^6$. For the experiment, carborundum strips were placed on the upper and lower surfaces at the $0.05\,c$ to ensure transition. The width of the strips was approximately $0.01\,c$. Numbers 60, 80, 120 and 180 indicate the carborundum grit sizes. The 60 strip was a extended from $0.05\,c$ on the upper surface to $0.05\,c$ on the lower surface.

For $\alpha = 0°$, the simulation using Spalart-Allmaras model gives $C_\mathrm{d} = 0.00817$ and $C_\mathrm{l} = -5.67 \times 10^{-10}$. The $k - \omega$ SST simulation for this angle of attack gave $C_\mathrm{d} = 0.00814$ and $C_\mathrm{l} = 2.07 \times 10^{-8}$. The values of $C_\mathrm{l}$ from these simulations are sufficiently close to zero for this symmetric airfoil. McCroskey (1987) surveyed the available 0012 experimental data and ranked them according to quality. His equation of fit to the best $C_\mathrm{d}$ data for an airfoil with $\alpha = 0°$ is:

$$C_\mathrm{d0} = 0.0017 + 0.91(\log Re)^{-2.58} \tag{25}$$





**Table 5.** First grid distance, growth ratio and $y^+$ information for this study simulations.

| Grid | First grid height | Growth ratio | Min. $y^+$ | Max. $y^+$ |
|---|---|---|---|---|
| Coarse grid | $1.08 \times 10^{-6}\ c$ | 1.32 | 0.0175 | 0.4080 |
| Medium grid | $5.4 \times 10^{-7}\ c$ | 1.15 | 0.0056 | 0.2024 |
| Fine grid | $2.7 \times 10^{-7}\ c$ | 1.07 | 0.0004 | 0.1013 |

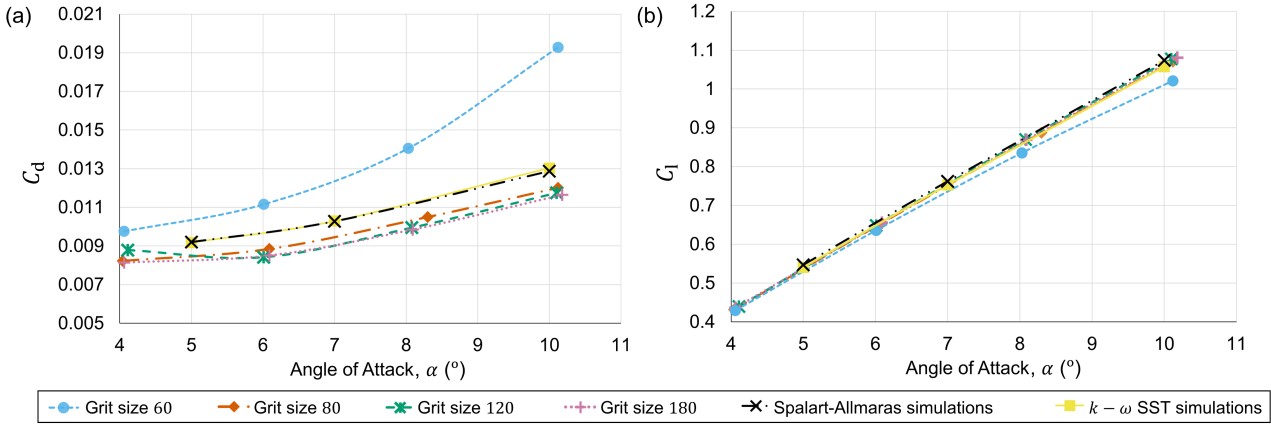

**Figure 7.** Comparison between $C_\mathrm{d}$ and $C_\mathrm{l}$ of the simulations and experimental results for NACA 0012, Re$\approx 6 \times 10^6$ and Ma$= 0.15$.

to an overall precision of about $\pm 0.0002$. At $Re = 6 \times 10^6$, $C_{\mathrm{d}_0}$ should be $= 0.00823 \pm 0.0002$, and our OpenFOAM simulation result lies within this range. The $C_\mathrm{d}$ values for all of Ladson's grit sizes except of 60 are within the range of $C_{\mathrm{d}0}$ from
McCroskey (1987), so we conclude that the data for the 60 grit are not reliable. We note that Rumsey compares his results with the Ladson (1988) experimental data for grit sizes other than 60. Finally, the present results show no substantial differences in the force coefficients between the two turbulence models.

## 5 Cascade simulations

This section describes the simulations of the flow through cascades at pitch angles of $90°$, $80°$ and $0°$ with varying $S/c$. For
$\beta = 90°$, a simulation was done for $\alpha_g = 0°$ to investigate a condition where there is no lift. $\alpha_g$, and the other angles and important symbols are defined in Fig. 2. The remaining cases have $\alpha_g = 10°$ or $4°$. Since there is no recommendation available for the appropriate domain size, a domain independence study for the cases at $\alpha_g = 10°$ with $\beta = 80°$ and $0°$ was undertaken. For the cascades at $\beta = 80°$ and $\alpha_g = 10°$, the domain with $A = 50$ and $B = 30$ was chosen, and the cascade simulations having $\beta = 0°$ and $\alpha_g = 10°$ required $A = 30$ and $B = 30$ to give similar results to the simulations with larger domains. The
$y^+$ values for the cascade are similar to those for the corresponding airfoil. For all cascade simulations in the following sections,





**Table 6.** Comparison between $C_{\mathrm{x}}$ from OpenFOAM to Eqs. (26) and (27).

| $S/c$ | $C_{\mathrm{x}}$ | $C_{\mathrm{x}}$ from Eq. (26) | $C_{\mathrm{x}}$ from Eq. (27) |
|---|---|---|---|
| 5 | 0.00822 | 0.00819 | 0.00744 |
| 10 | 0.00818 | 0.00815 | 0.00778 |
| 20 | 0.00817 | 0.00813 | 0.00795 |
| 40 | 0.00817 | 0.00812 | 0.00802 |

the maximum value of $y^+$ is $\simeq 0.1$. Note that $y-$extent of the domain is $S/c$ rather than $2A$. The simulation results with their initial interpretations are presented in the following subsections.

Hybrid grids were used to restrict the number of cells, with the structured meshes concentrated on the bodies and the body wakes. Unstructured grids affect the smoothness of the vorticity, so calculating the terms containing the vorticity is more
accurate when the wake is covered by the structured grid. For $\beta = 0°$, the wakes cross the top and bottom boundaries of the computational domain. In this case, it was more accurate to determine the wake quantities when the wake is in the middle of $S_2$.

### 5.1 $\alpha_g = 0°, \beta = 90°$ and varying $S/c$

Since the NACA 0012 airfoil is symmetrical, it generates no lift when $\alpha = 0°$. Thus, $\widetilde{V}_2 = 0$ and Eq. (14) reduces to:

$$\frac{F_x}{\rho} \approx \int\limits_{wake} \left( U_2 + \frac{\overline{U}_2 - U_1}{2} \right) \left( \overline{U}_2 - U_2 \right) \mathrm{d}y. \tag{26}$$

The computed values of $-\widetilde{V}_b \Gamma$ (as expected) and $\int\limits_{S_2} v_2^2 \mathrm{d}y/2$ were negligible as was $C_{\mathrm{y}}$. As mentioned above, our results for $C_{\mathrm{d0}}$ lie within the range given by McCroskey (1987) for $Re = 6 \times 10^6$.

Since $\overline{U}_2 \xrightarrow[S/c \to \infty]{} U_1$, Eq. (26) reduces to

$$\frac{F_x}{\rho} \xrightarrow[S/c \to \infty]{} \int\limits_{wake} U_2(U_1 - U_2)\mathrm{d}y. \tag{27}$$

This equation is the conventional drag equation for an isolated, non-lifting body, which is derived in most fundamental fluid mechanics books using the continuity and conventional momentum equations (e.g., Anderson, 2017, Eq. (2.84)). Table 6 compares the OpenFOAM results with Eq. (26) and Eq. (27) for these four spacing ratios. As expected, Eq. (27) becomes more accurate as $S/c$ increases, but blockage is important at the lower $S/c$ even though the magnitude of $C_{\mathrm{x}}$ does not change significantly.
We repeat the statement made in the Introduction that BEM does not contain a term representing the viscous drag which is equivalent to assuming $C_{\mathrm{x}} = 0$ for this cascade case.



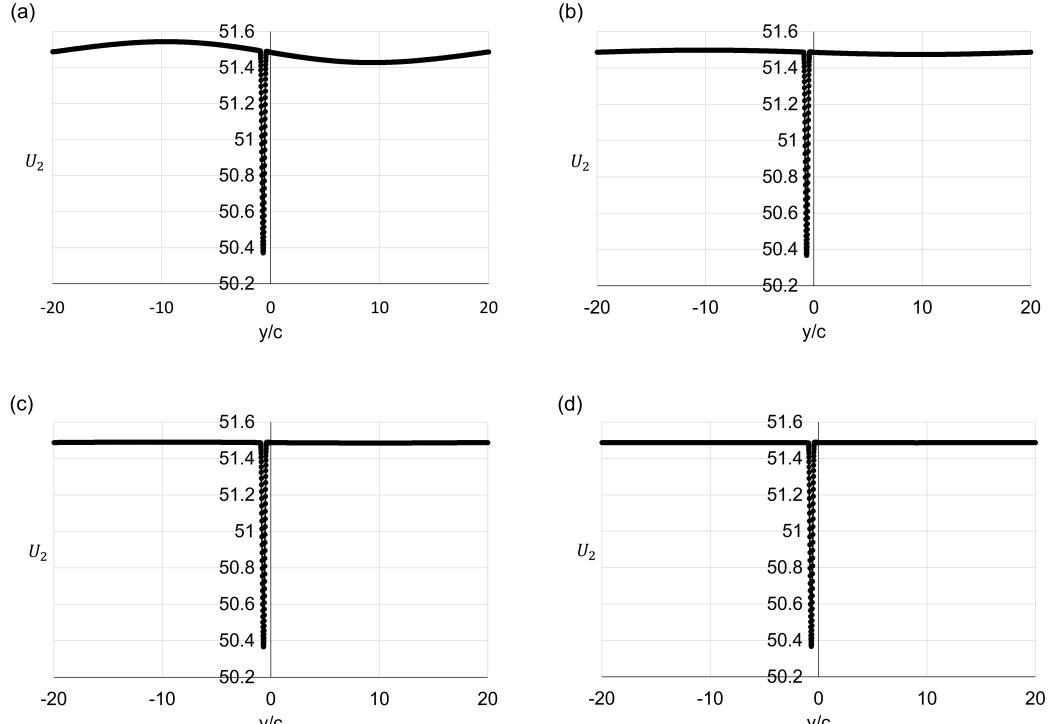

**Figure 8.** $U_2$ at the outlet for cases with $\alpha_g = 10°$, $\beta = 80°$, $S/c = 40$, $B = 30$ and different values for $A$; (a) $A = 20$, (b) $A = 30$, (c) $A = 40$ and (d) $A = 50$.

## 5.2  $\alpha_g = 10°, \beta = 80°$ and varying $S/c$

This section discusses simulations of the flow through cascades of NACA 0012 airfoils at $\alpha_g = 10°$, $\beta = 80°$ and varying $S/c$ (5, 10, 20, 40). As mentioned in the Introduction, this high value of $\beta$ corresponds to a low tip speed ratio. First, an appropriate

domain size must be determined using a domain independence study. We started with $S/c = 40$ and the domain size used for single airfoil simulations, $A = 20$ and $B = 30$. For fixed $A$ and $B$, the larger the value of $S/c$, the smaller was $F_x$, so the impulse equation for the $x$ component of the force is more sensitive to error. Therefore, we undertook the domain independence study for the highest $S/c = 40$. First, the outlet was set at $B = 30$ and $A$ was varied. Figure 8 shows the $U_2$ profile for these cases. It is reasonable to require $A$ to be large enough to ensure the validity of the slender flow approximation which requires

that $U_2$ is constant outside the wake. There is, however, vorticity outside the wake at the outlet for $A = 20, 30$ and $40$. Since we derived Eq. (14) from Eq. (12) by assuming that the flow is fully developed and $U_2$ is constant outside the wake at the outlet, we determined the difference in $F_x$ between these two equations. Note that we assumed $U_2$ outside the wake to be $\overline{U}_2$ in evaluating Eq. (14). Based on Table 7, which compares the value of Eqs. (12) and (14), the domain with $A = 50$ has an error of less than 1 % between these two equations. The percent error between a real value ($A_{real}$) and its approximation ($A_{approx}$)

is defined as $100\% \times |A_{real} - A_{approx}|/A_{real}$. Therefore, we temporarily set $A = 50$ and continued the domain independence





study by changing $B$. Table 7 also shows that changing $B$ affects $U_2$ outside the wake. Reducing $B$ from 30 to 20 caused an error of 2.52 % between Eqs. (12) and (14), and increasing $B$ had no adverse effect on the accuracy of Eq. (14). Therefore, an appropriate domain size for cascade simulations is $A = 50$ and $B = 30$. Case 7 in Table 7 implies that even if we started the domain independence study by fixing the inlet and changing the outlet, eventually, we would end up with the chosen domain.

**Table 7.** Comparison between $C_x$ derived from Eqs. (12) and (14) for cascades with $\alpha_g = 10°$, $S/c = 40$, and varying $A$ and $B$ at $x = Bc$.

| Case No. | $A$ | $B$ | $C_x$ from Eq. (12) | $C_x$ from Eq. (14) | % error |
|---|---|---|---|---|---|
| 1 | 20 | 30 | 0.02017 | 0.01351 | 33.02 % |
| 2 | 30 | 30 | 0.01896 | 0.01763 | 7.06 % |
| 3 | 40 | 30 | 0.01866 | 0.01843 | 1.23 % |
| 4 | 50 | 30 | 0.01859 | 0.01860 | 0.03 % |
| 5 | 50 | 20 | 0.01866 | 0.01913 | 2.52 % |
| 6 | 50 | 40 | 0.01859 | 0.01851 | 0.43 % |
| 7 | 40 | 40 | 0.01872 | 0.01838 | 1.79 % |

Table 8 indicates that for all cases in Table 7 the differences in $C_x$ and $C_y$ for different domain sizes are less than 1 %, so the force components are not sensitive to changing domain sizes for the domains in Table 7. As a result, we continued simulations of cascades with $\alpha_g = 10°$ and varying $S/c$ with the domain size of $A = 50$ and $B = 30$. Figure 9 shows the grid

**Table 8.** Comparison between $C_x$ and $C_y$ calculated by OpenFOAM for cascades with $\alpha_g = 10°$, $S/c = 40$, and varying $A$ and $B$.

| Case No. | $A$ | $B$ | $C_x$ | % error of $C_x$ with case 4 | $C_y$ | % error of $C_y$ with case 4 |
|---|---|---|---|---|---|---|
| 1 | 20 | 30 | 0.01858 | 0.16 % | 1.03922 | 0.005 % |
| 2 | 30 | 30 | 0.01855 | 0 % | 1.03928 | 0.001 % |
| 3 | 40 | 30 | 0.01855 | 0 % | 1.03927 | 0 % |
| 4 | 50 | 30 | 0.01855 | 0 % | 1.03927 | 0 % |
| 5 | 50 | 20 | 0.01857 | 0.11 % | 1.03912 | 0.01 % |
| 6 | 50 | 40 | 0.01855 | 0 % | 1.03927 | 0 % |
| 7 | 40 | 40 | 0.01855 | 0 % | 1.03928 | 0.001 % |

independence study for the case with $A = 50$ and $B = 30$ and $S/c = 40$. $GCI_{23}/\mathfrak{r}^p GCI_{12}$ (Roache, 1994) for both $C_x$ and $C_y$ is approximately 1. Also, $C_x$ and $C_y$ for infinite numbers of cells are 0.01859 and 1.03929 with the error band of 0.03 % and

0.002 % , so around 2 million cells gave accurate results.

$C_x$ from OpenFOAM simulations, Eqs. (10), (12), (14) and (18) for different values of $S/c$ are compared in Table 9. These three equations agree with the direct determination of the body forces with less than 1 % error. This table also indicates that





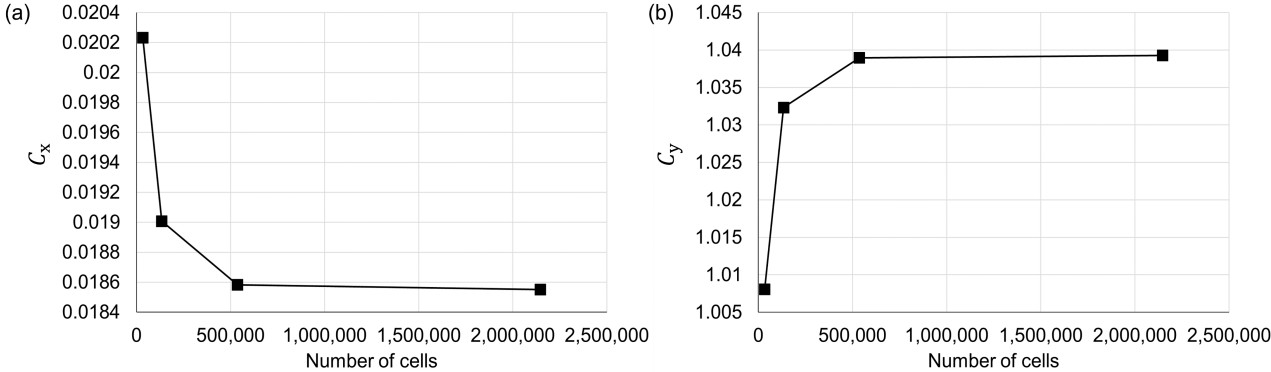

**Figure 9.** Grid convergence study of (a) $C_x$ and (b) $C_y$ for the cascade with $\alpha_g = 10°$ and $S/c = 40$.

**Table 9.** Comparison of $C_x$ from OpenFOAM and Eqs. (10), (12), (14), (18), and comparison between $C_y$ from OpenFOAM with Eq. (20) for cascades at $\alpha_g = 10°$, $\beta = 80°$ and varying $S/c$.

| $S/c$ | $C_x$ by OpenFOAM | $C_x$ from Eq. (10) | $C_x$ from Eq. (12) | $C_x$ from Eq. (14) | $C_x$ from Eq. (18) | Max. % error | $C_y$ by OpenFOAM | $C_x$ from Eq. (20) | % error |
|---|---|---|---|---|---|---|---|---|---|
| 5 | 0.04366 | 0.04366 | 0.04368 | 0.04359 | 0.04359 | 0.15 % | 0.81672 | 0.81669 | 0.003 % |
| 10 | 0.03285 | 0.03286 | 0.03286 | 0.03284 | 0.03283 | 0.08 % | 0.93512 | 0.93511 | 0.002 % |
| 20 | 0.02407 | 0.02408 | 0.02407 | 0.02406 | 0.02407 | 0.03 % | 1.00303 | 1.00302 | 0.001 % |
| 40 | 0.01855 | 0.01856 | 0.01859 | 0.01860 | 0.01856 | 0.26 % | 1.03927 | 1.03927 | 0.00007 % |

the values of $C_y$ from the OpenFOAM simulations and Eq. (20) for the four values of $S/c$ are in good agreement. Figure 10 (a) represents $C_x$ from Eq. (18) and its terms for cascades at $\alpha_g = 10°$, $\beta = 80°$ and having different $S/c$. As $S/c$ increases,

$F_x$ and the circulation contribution to $F_x$ decreases. Parts (a) and (b) of this figure indicate that with the decrease of $S/c$, the lift vector turns more in the $y-$direction to make $F_x$ larger and $F_y$ smaller. As $S/c$ increases, the circulation contribution (blue squares) tends to zero and the blockage effect (green diamonds) asymptotes to $C_d$ for an airfoil the $\alpha = 10°$. For example, for $S/c = 40$, the normalized first term in (18) is 0.01181 and close to $C_d$ for the single airfoil with $\alpha = 10°$, which is 0.01287. The order of magnitude of the normalized non-linear term in Eq. (18) for all values of $S/c$ is $10^{-5}$ or less, so it is negligible,

and it is not shown in Fig. 10. This figure also shows the monotonic increase in $F_y$ with increasing $S/c$, which agrees with the theoretical results of Baddoo and Ayton (2018). For $S/c = 40$, $C_y = 1.03927$, and as $S/c$ increases, this value tends to $C_l$ of the airfoil with $\alpha = 10°$, which is 1.07379. Table 10 presents the normalized values of $\overline{U}_2$ and $\widetilde{V}_b$ for the case with the lowest $S/c = 5$ as an example, at different ($x = 5\,c$, $10\,c$, $20\,c$, and $30\,c$). The $x-$component of the velocity has big changes outside the wake at $x = c$ and $2\,c$, so the slender flow approximation is not valid as indicated by "-". $\widetilde{V}_b$ remains approximately

constant, which indicates that only the vorticity bound to the blade contributes to $\Gamma$, as a consequence of Eq. (19). Table 10 also shows the normalized terms of Eq. (14) at different locations for $S/c = 5$. It is observed that the farther from the body,





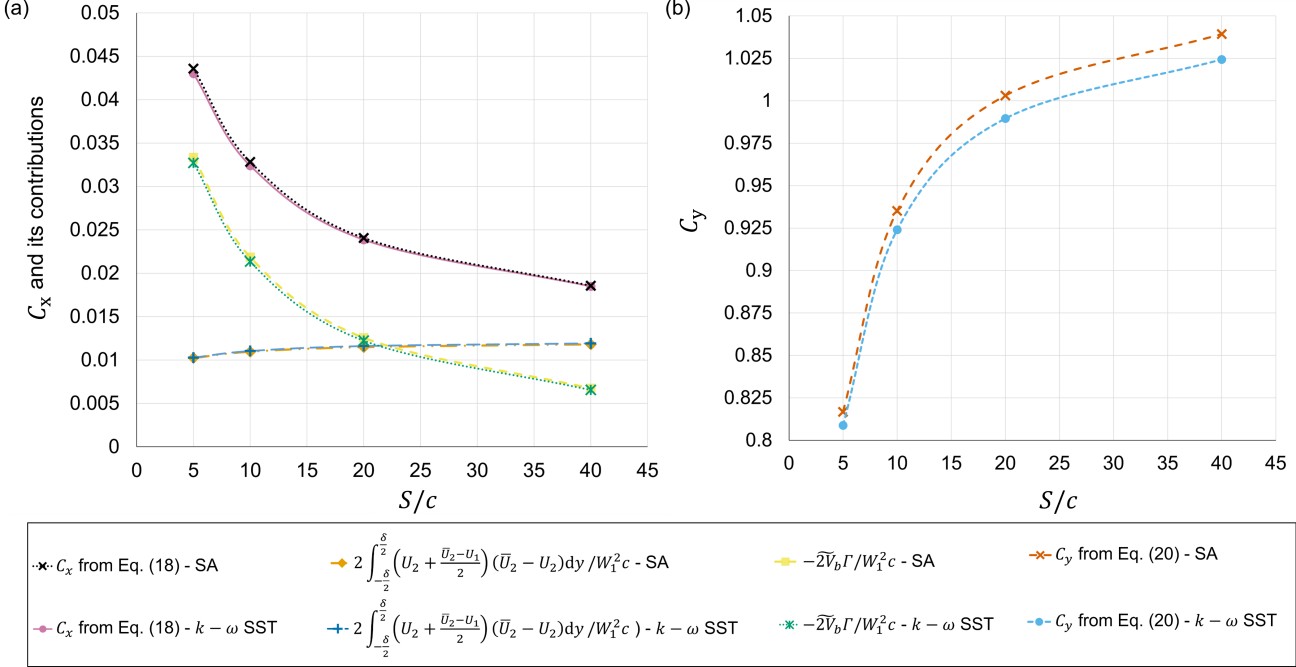

**Figure 10.** (a) $C_{\mathrm{x}}$ and terms of Eq. (18), (b) $C_{\mathrm{y}}$ from Eq. (20) cascades at $\alpha_g = 10°$ and $\beta = 80°$; vs $S/c$ simulated using Spalart-Allmaras and $k - \omega$ SST models evaluated at $x = 30\,c$.

the smaller the non-linear term and the more accurate the value of Eq. (14). The table clearly shows, however, that there is no significant interchange of force between the various components of the force balance.

### 5.3   $\alpha_g = 10°, \beta = 0°$ and varying $S/c$

Figure 11 presents $C_{\mathrm{x}}$, and $C_{\mathrm{y}}$ for cascades at $\alpha_g = 10°$, $\beta = 0°$ and different $S/c$. This $\beta$ is appropriate to the tip region of wind turbine blades at high tip speed ratio. Both $C_{\mathrm{x}}$ and $C_{\mathrm{y}}$ decrease with increase of $S/c$. Table 11 compares $C_{\mathrm{x}}$ derived from OpenFOAM simulations with Eqs. (10) and (12) and the direct determination of $C_{\mathrm{y}}$ with Eq. (20) for cascades at $\alpha_g = 10°$, $\beta = 0°$ and varying $S/c$. Equations (10), (12) and (20) for all cases in Table 11 were calculated at $x$, between $25\,c - 40\,c$. The sections are shown in Fig. 12.

This figure also shows the excellent behaviour of the periodic boundary conditions, which allow the wake to continually pass through the top and bottom boundaries. As it can be seen in part (a) of Fig. 12, at $S/c = 5$, there is a deflection in the wake, which comes from the above body, caused by the airfoil shown in the figure.

### 5.4   $\alpha_g = 4°, \beta = 0°$ and varying $S/c$

This condition was chosen to mimic the high thrust region of wind turbine operation where the blade element $\alpha$ is reduced
below that giving optimal lift:drag. The wake for the cascade is shown in Fig. 13. The wake is now approaching $S_2$ at a very





**Table 10.** Velocities and momentum balance terms for $S_2$ at different distances from the airfoil. $\alpha_g = 10°$, $\beta = 80°$ and $S/c = 5$. The OpenFOAM simulation result for $C_x$ is 0.04366.

| | $x = c$ | $x = 2c$ | $x = 5c$ | $x = 10c$ | $x = 20c$ | $x = 30c$ |
|---|---|---|---|---|---|---|
| $\overline{U}_2/W_1$ | - | - | 1.00101 | 1.00105 | 1.00104 | 1.00104 |
| $2\widetilde{V}_b/W_1$ | $-0.08168$ | $-0.08167$ | $-0.08167$ | $-0.08167$ | $-0.08167$ | $-0.08167$ |
| $(U_1^2 - \overline{U}_2^2)S/W_1^2 c$ | - | - | $-0.01006$ | $-0.01049$ | $-0.01042$ | $-0.01038$ |
| $2\int\limits_{wake}(\overline{U}_2^2 - U_2^2)\mathrm{d}y/W_1^2 c$ | - | - | 0.02098 | 0.02075 | 0.02066 | 0.02062 |
| $-2\widetilde{V}_b\Gamma/W_1^2 c$ | 0.03335 | $-0.03335$ | 0.03335 | 0.03335 | 0.03335 | 0.03335 |
| $\int\limits_{S_2} v_2^2\mathrm{d}y/W_1^2 c$ | 2.74525 | 0.18241 | $1.49 \times 10^{-6}$ | $9.50 \times 10^{-7}$ | $6.25 \times 10^{-7}$ | $4.92 \times 10^{-7}$ |
| $2\int\limits_{wake}\left(U_2 + \frac{\overline{U}_2-U_1}{2}\right)\left(\overline{U}_2 - U_2\right)\mathrm{d}y/W_1^2 c$ | - | - | 0.01027 | 0.01023 | 0.01024 | 0.01024 |
| $2F_x/\rho W_1^2 c$ (from Eq. (12)) | 0.04367 | 0.04361 | 0.04367 | 0.04368 | 0.04367 | 0.04368 |
| $2F_x/\rho W_1^2 c$ (from Eq. (14)) | - | - | 0.04427 | 0.04361 | 0.04359 | 0.04359 |
| $2F_x/\rho W_1^2 c$ (from Eq. (18)) | - | - | 0.04362 | 0.04358 | 0.04359 | 0.04359 |

**Table 11.** Comparison of $C_x$ from OpenFOAM and Eq. (12), and comparison between $C_y$ from OpenFOAM and Eq. (20) for cascades at $\alpha_g = 10°$, $\beta = 0°$ and varying $S/c$.

| $S/c$ | $x$ $(d)$ [m] | $C_x$ by OpenFOAM | $C_x$ from Eq. (10) | $C_x$ from Eq. (12) | Max % error | $C_y$ by OpenFOAM | $C_x$ from Eq. (20) | % error |
|---|---|---|---|---|---|---|---|---|
| 5 | 3.9 $c$ ( 25 $c$) | 1.16229 | 1.16229 | 1.16264 | 0.03 % | 0.18142 | 0.18142 | 0 % |
| 10 | 4.9 $c$ (30 $c$) | 1.10320 | 1.10320 | 1.10270 | 0.05 % | 0.17690 | 0.17690 | 0 % |
| 20 | 6.7 $c$ (41 $c$) | 1.08226 | 1.08226 | 1.08272 | 0.04 % | 0.17589 | 0.17589 | 0 % |
| 40 | 7 $c$ (41 $c$) | 1.07375 | 1.07375 | 1.07700 | 0.30 % | 0.17569 | 0.17569 | 0 % |

shallow angle and we found that no analysis of the force equations gave unambiguous results. The wakes of proximate bodies are clearly interfering with each other. The terms of the impulse equation $20c$ far from the body along the wake are presented in table 12, which is elaborated in Subsection 6.





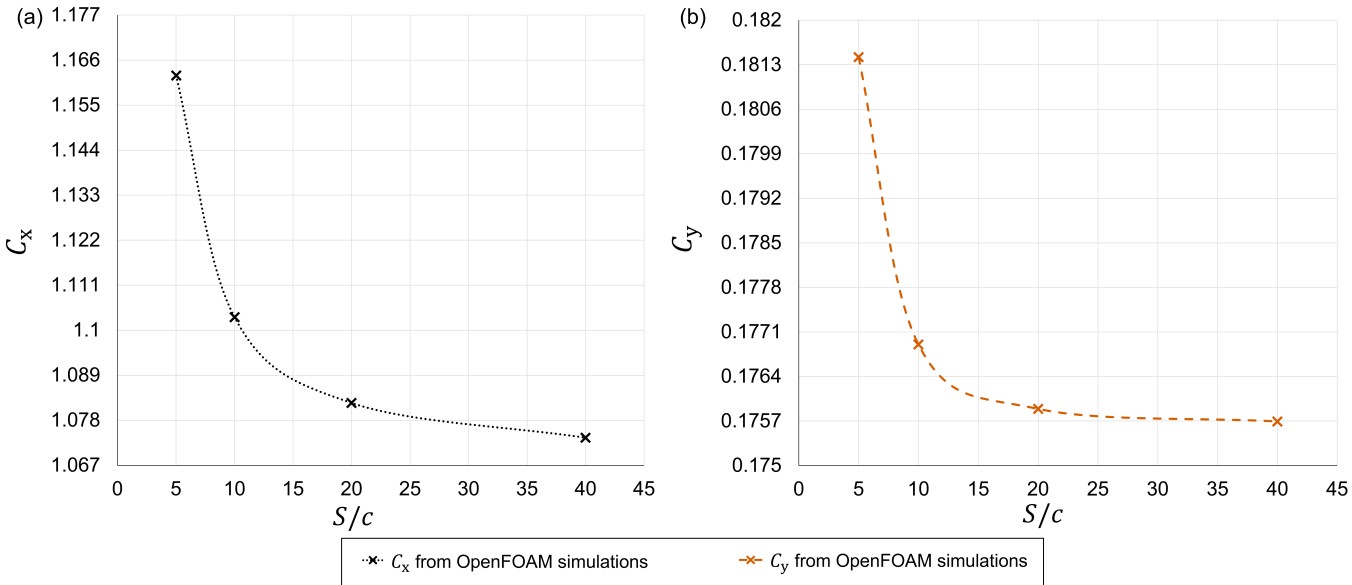

**Figure 11.** (a) $C_{\mathrm{x}}$, and (b) $C_{\mathrm{y}}$ for cascades at $\alpha_g = 10°$, $\beta = 0°$ and varying $S/c$.

# 6 Implications for blade element modeling

## 6.1 Wake vorticity and element drag

Having replaced the wake vorticity term in the $F_x$ equation to arrive at Eq. (27), we return to Eq. (12), which can be rewritten as

$$\frac{F_x}{\rho} = -\widetilde{V}_b\Gamma + \frac{1}{2}U_1^2 S - \int_{S_2} \frac{1}{2}U^2\mathrm{d}y - \int_{S_2} yU\Omega_z\mathrm{d}y + \frac{1}{2}\int_{S_2} v_2^2\mathrm{d}y. \tag{28}$$

The quadratic terms in $U$, $U_2$, and $v_2$ were found to be negligible for all cases and so are not given in Table 12 which lists the remaining terms for the simulations discussed in this section. Similarly, the quadratic terms for $F_y$ in Eq. (20) were also negligible.

We note again that the BEM version of Eq. (28) derived by Limacher and Wood (2021) contains only the equivalent of the first term on the right, whereas the $y-$direction BEM equation, corresponding to $F_y = \rho U_1\Gamma$, is complete, at least for circumferentially-uniform flow. The wake vorticity term in the second last column balances $F_x$ completely when it is due to the body drag at $\beta = 90°$ but still represents 7 % of $F_x$ in the third row when $F_x$ is dominated by the circulation term. The contribution increases to approximately 27 % of $F_x$ for the fourth case. Equally importantly for the third row is that $U_1\Gamma$ balances $F_y/\rho$ even when $F_y$ has a contribution from the drag. In other words, the circulation in the wake is partly determined by the blade element drag.

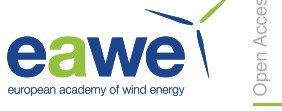

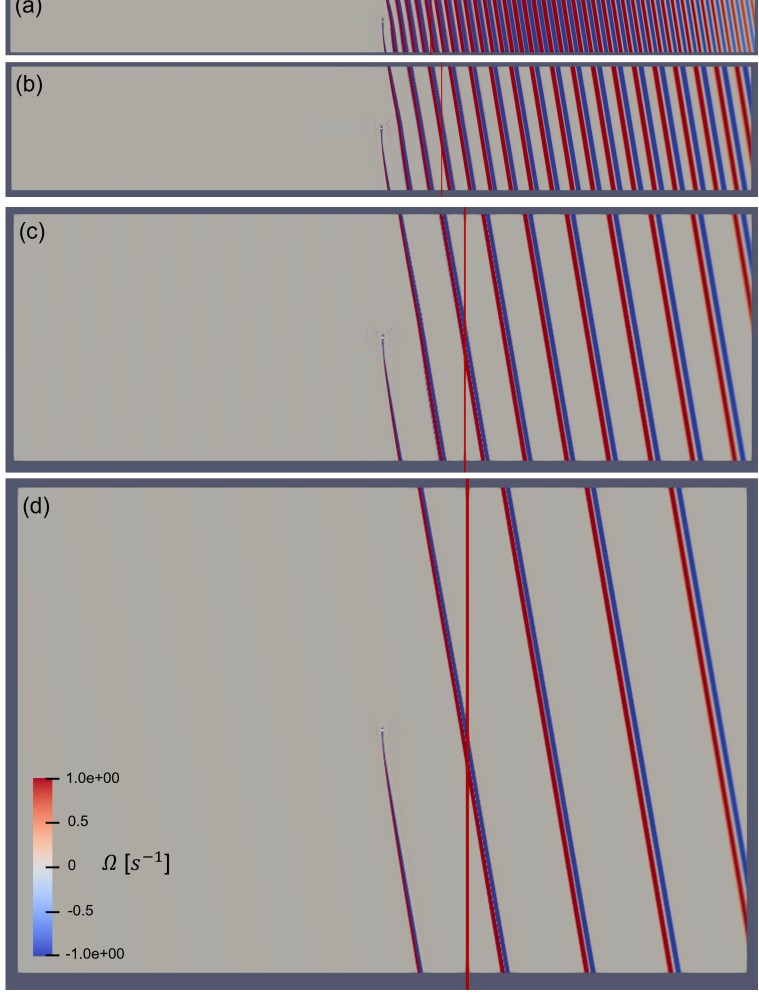

**Figure 12.** Vorticity contours, rescaled between −1 and 1, for the cascade simulations at $\beta = 0°$, $\alpha_g = 10°$ and (a) $S/c = 5$, (b) $S/c = 10$, (c) $S/c = 20$, (d) $S/c = 40$. The regions shown are the entire computational domains with the airfoil body in the centre of each part of the figure. The red lines show the location of $S_2$ used to calculate the forces in Table 11.

The importance of the wake vorticity term in Table 12 suggests the need to include it in BEM analysis. Clearly, it must be

related to the element drag, $D$. Dropping the negligible terms in Eq. (28) gives

$$\frac{F_x}{\rho} \approx -\widetilde{V}_b\Gamma - \int_{S_2} yU\Omega_z \mathrm{d}y \qquad (29)$$





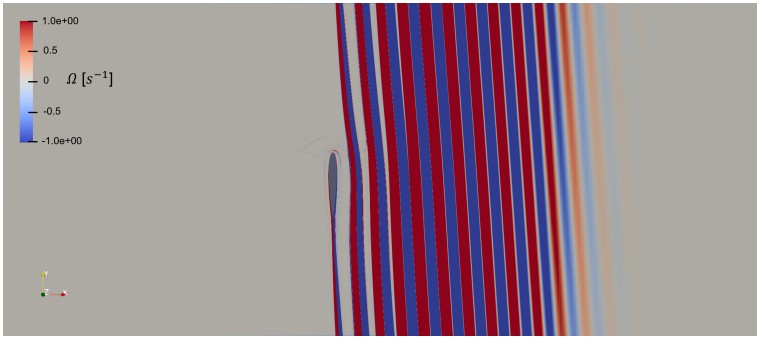

**Figure 13.** Vorticity contours for the cascade simulations at $\beta = 0°$, $\alpha_g = 4°$, and $S/c = 5$. The vorticity is rescaled between $-1$ and $1$.

**Table 12.** Force components, lift and drag for cascade bodies with $S/c = 5$: case 1, $\beta = 90°$, $\alpha_g = 0°$; case 2, $\beta = 80°$, $\alpha_g = 10°$; case 3, $\beta = 0°$, $\alpha_g = 10°$; and case 4, $\beta = 0°$, $\alpha_g = 4°$. The units of all quantities are $[m^3 s^{-2}]$.

| Case | $\frac{F_x}{\rho}$ | $\frac{F_y}{\rho}$ | $\frac{L}{\rho}\sin\theta$ | $\frac{L}{\rho}\cos\theta$ | $\frac{D}{\rho}\sin\theta$ | $\frac{D}{\rho}\cos\theta$ | $\frac{D}{\rho\cos\theta}$ | $-\widetilde{V}_b\Gamma$ | $-\int\limits_{S_2} yU\Omega_z\mathrm{d}y$ | $U_1\Gamma$ |
|------|------|------|------|------|------|------|------|------|------|------|
| 1 | 10.89 | 0 | 0 | 0 | 0 | 10.89 | 10.89 | 0 | 10.96 | 0 |
| 2 | 57.88 | 1082.23 | 44.22 | 1082.79 | 0.56 | 13.66 | 13.69 | 44.19 | 13.78 | 1082.21 |
| 3 | 1540.61 | 240.40 | 1537.75 | 257.50 | 17.10 | 2.86 | 104.97 | 1434.62 | 105.55 | 240.23 |
| 4 | 608.28 | 30.50 | 607.50 | 41.79 | 11.29 | 0.78 | 164.87 | 445.31 | 163.40 | 30.63 |

and $F_y$ is given by Eq. (21). Combining the two equations with Eqs. (4) and (5), and using $\tan\theta = -\widetilde{V}_b/U_1$ immediately leads to

$$-\int\limits_{S_2} yU\Omega_z\mathrm{d}y \approx \frac{D}{\rho\cos\theta}. \tag{30}$$

Table 12 shows that $D/(\rho\cos\theta)$ is an accurate approximation for the wake vorticity term that should be easy to include in BEM analysis. This is the major result of our analysis.

### 6.2 The accuracy of the airfoil assumption

For $S/c = 5, 10, 20$ and $40$, using Eq. (3) and the equation of velocity at the body, we calculated $C_l$ and $C_d$ and compared these values to the experimental data available for NACA 0012 airfoil from Ladson (1988) in Fig. 14. Our simulations show the
airfoil assumption is reasonably accurate but conservative even for $\beta = 0$. Also, Fig. 15 compares the lift to drag ratios of the simulations with the NACA 0012 experimental data. For $\beta = 80°$, $L/D$ decreases more rapidly than the airfoil $L/D$ as $S/c$ decreases. For the cascade at $\beta = 0°$, $L/D$ increases with decrease in the spacing ratio and $\alpha$.

Figure 16 compares the pressure coefficient, $C_p$, the skin friction coefficient, $C_f$ of the cascade at $\beta = 0°$, $S/c = 5$ and $\alpha_g = 10°$ ($\alpha = 9.5057°$) with a single airfoil at the same $\alpha$. Both $C_p$ and $C_f$ are larger for the cascade than the single airfoil





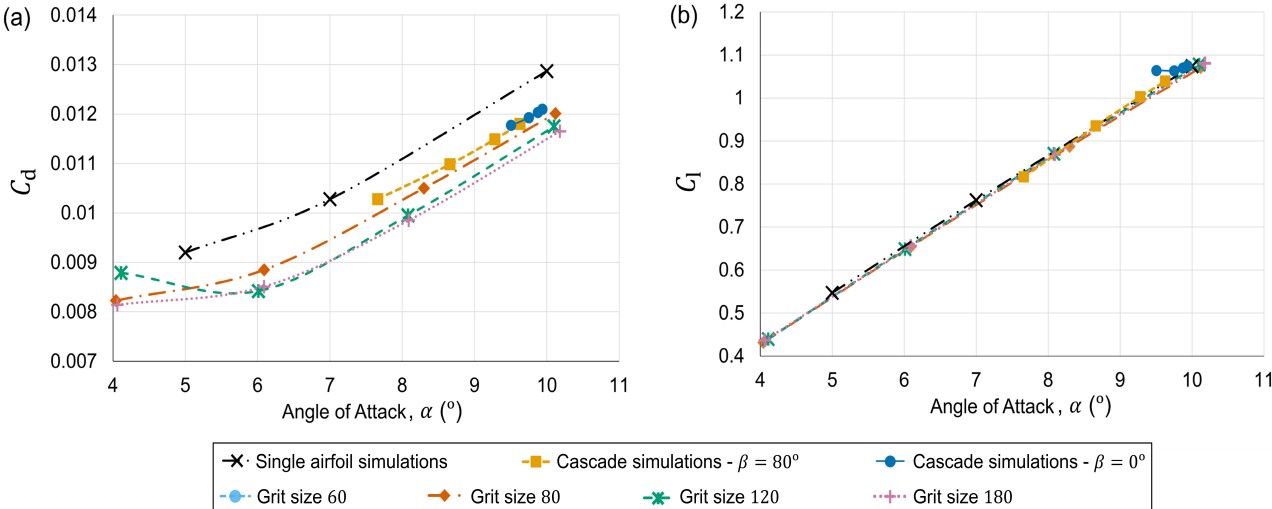

**Figure 14.** Assessing the airfoil assumption using a comparison between the simulation results for different values of $S/c$ given in the legend and experimental results from Ladson (1988). (a) $C_d$ vs $\alpha$. (b) $C_l$ vs $\alpha$. "Grit" refers to the roughness used on the airfoil model to induce transition.

near the leading edge on the upper surface. Since the relative increase of $C_p$ is larger than the relative increase of $C_f$, the lift to drag ratio is greater for the cascade than the single airfoil, which is also can be seen in Fig. 15.

Hu et al. (2006) argue that the Coriolis and centrifugal forces have substantial effects on delaying the separation of the boundary layer of a wind turbine blade compared with an airfoil. This phenomenon is called "stall delay". Figure 16 suggests that stall delay can occur simply because of increased solidity. The adverse pressure gradient at the upper surface of a wind

turbine blade decreases compared to an airfoil with the same angle of attack, so the cascade flow separates at an angle of attack larger than the ones that a single airfoil is being to be stall away. It must be noted, however, that Fig. 16 shows the extreme case in terms of the cascade solidity.

## 7   Conclusion

This study describes simulations of flows over cascades of lifting bodies and their application to horizontal-axis wind turbines.

A cascade is a two-dimensional unfolding of an axisymmetric rotor which allows an investigation of some aspects of finite rotor solidity and the equations for the axial and normal momentum balances in the wakes. Cascade flows do not have the equivalent of the vorticity shed by blades due to radial gradients of the bound vorticity, but they do contain the "wake vorticity" associated with blade element drag. The equations used in conventional blade-element/momentum analysis ignore the wake vorticity: the main outcome of this study was to show its importance and to derive an expression for it - Eq. (30) - that may

well be useful for wind turbine analysis.

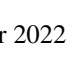



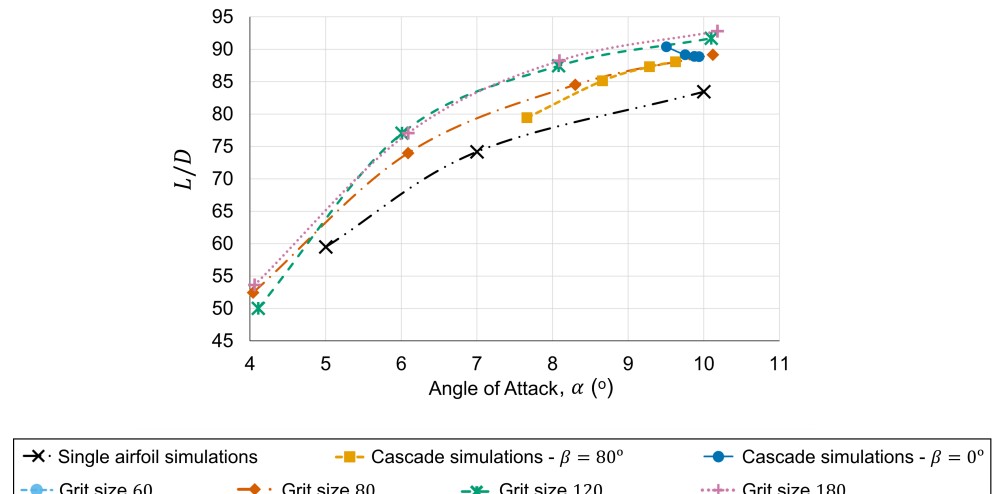

**Figure 15.** Lift to drag ratio for single airfoils and cascades simulations and experimental data from Ladson (1988).

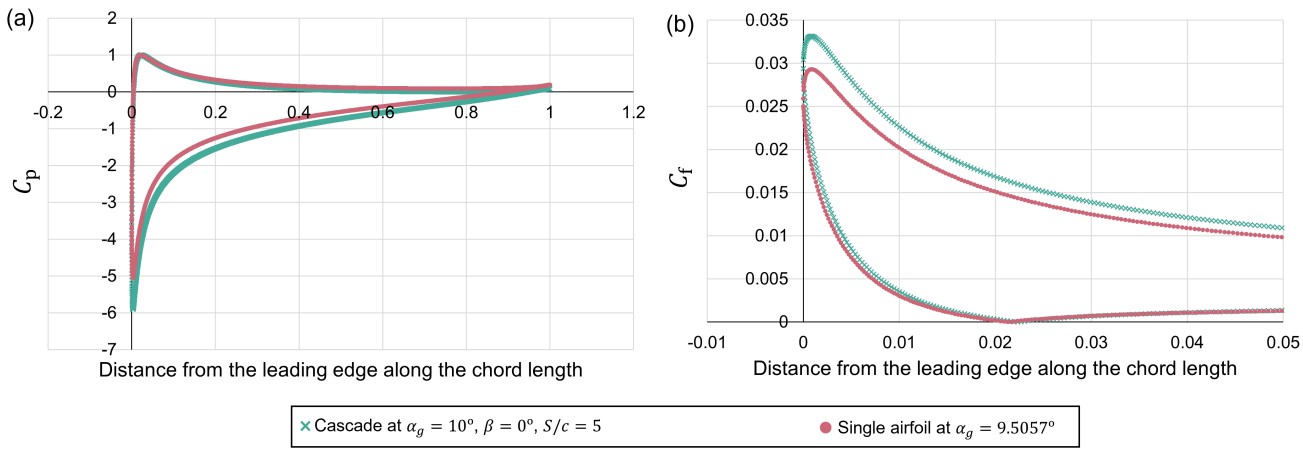

**Figure 16.** Comparison of (a) $C_\mathrm{p}$, and (b) $C_\mathrm{f}$ between cascade at $\alpha_g = 10°$, $\beta = 0°$ and $S/c = 5$ and single airfoil at $\alpha_g = 9.5057°$. The $x$-axis of part (b) is zoomed to make the difference clear.

Equation (30) and the other results were analyzed using the conventional equations for streamwise and normal momentum as well as the impulse version of the former which identifies the wake vorticity term. The cascade equations were shown to reduce to the conventional equation for the drag of a nonlifting body as the spacing ratio, defined as the body spacing divided by the body chord, goes to infinity.

Because the momentum equations are nonlinear, they have contributions from the nonuniformity of the wake velocities. These were found to be negligible in all cases, implying that circumferential nonuniformity in wind turbine wakes is caused overwhelmingly by the shed vorticity. The normal momentum equation contains no terms corresponding to the wake vorticity





for the axial momentum, whereas the element force equations show that drag contributes to the normal force in general. This means that the circulation in cascade wakes, and, by implication wind turbine wakes, is determined partly by blade element

drag. This is contrary to use of only the element lift to determine the circulation in many BEM analyses.

One of the most important, and least tested assumptions in blade element theory is that the elements behave as airfoils. Cascade simulations allow a direct test of the assumption, which we found to be conservative: the lift:drag ratio of cascade elements was always greater than that of the corresponding airfoils. For the smallest spacing ratio considered, this conservatism was related to the solidity-induced delay in separation from the element.

*Author contributions.*  The study was conceived jointly. The CFD simulations were done by NG under the supervision of DW. The article was written jointly.

*Competing interests.*  The authors declare no competing interests.

*Acknowledgements.*  This work is supported by NSERC Discovery Grant RGPIN/04886-2017 and the Schulich endowment to the University of Calgary.





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
