# Peer review of "Investigating Horizontal Axis Wind Turbine Aerodynamics Using Cascade Flows"

_Wind Energy Science, 2022_

## Referee Comment (RC1)

Review: Manuscript WES-2022-76

Title: Investigating Horizontal Axis Wind Turbine Aerodynamics Using Cascade Flows

Authors: Narges Golmirzaee and David H. Wood

**1. General**

a) Summary as understood by the referee

The manuscript is understood as an investigation of the influence of the neighbouring of the other blades to a specific one. Apart from that a lot of addition items are presented.

b) Literature to be added

It seems that "cascade effects" are (at least) investigated in the following sources.

R. E. Wilson, P.B.S. Lissaman and S.N. Walker, in Aerodynamic Performance of Wind Turbines, Oregon State University, Corvallis, Oregon, USA, June, 1977, Section 2.7

N. Scholz, Aerodynamik der Schauffelgitter, 2 Bände, Karlsruhe, Germany, Verlag G. Braun, 1965

(probably English equivalents are available from literature about many-bladed turbo-machines

The authors are encouraged to improve their presentation of the state of the art

c) A remark about notation

"≈" is not a well-defined symbol. Please replace it with more accurate and scientific one or explain to what extent of accuracy "≈" refers to. Use of this symbol make it rather difficult to follow derivations.

d) A *complete* list of used symbols including their meaning might be helpful

**2. Specific**

Page 2 Fig 1: Introduce (consistent) coordinates

Page 3 Eq (6). Usually an unperturbed velocity appears in the KH-Theorem. Please explain why $W_{rel}$ can be used

Page 4, line 75 (and earler) pleas gibe a clear definition (in term of an equation) what "wake voriticity" really means

Page 5, section 2: It may be useful to explain - in short words - how the authors define the difference of (linear) "momentum" and "impulse".

Page 8, section 3: Which version of OpenFOAM was used?

Page 9, ff section 3.1. To the referee's opinion, the size of the computational domain (measured in terms of A and B) is not large enough. 100 c(hords) is common practise.

As a consequence, presentation of cases umber 1 -6 are more or less not very important.

Page 12, line 255: what do you mean by "sufficiently close"?, Please add some exp. Values in Figure 7.

Page 19, Table 11: please replace "0 %" by "< … $10^{-n}$"

Page 20, line 362. "In other words … the circulation in the wake . . . by the blade element drag". Circulation is a concept form inviscid flow. The authors should state (maybe already in the introduction) how they extend its definition and meaning to inviscid flow.

Page 23, lines 382 ff. "stall delay . . . increased solidity" Interesting idea, but I'm not sure if there is enough support for this hypothesis.

Page 23, section 7 "Conclusion".

Please distinguish between a "summary" and real "conclusions"